# Three-dimensional characterization of developing and adult ocular vasculature in mice using *in toto* clearing

Marie Darche [1,2], Anna Verschueren [1,2,7], Morgane Belle[2,7], Leyna Boucherit[1,2], Stéphane Fouquet[2], José Alain Sahel[2,3,4], Alain Chédotal [2], Ilaria Cascone[5,6] & Michel Paques [1,2 ✉]

The ocular vasculature is critically involved in many blinding diseases and is also a popular research model for the exploration of developmental and pathological angiogenesis. The development of ocular vessels is a complex, finely orchestrated sequence of events, involving spatial and temporal coordination of hyaloid, choroidal and retinal networks. Comprehensive studies of the tridimensional dynamics of microvascular remodeling are limited by the fact that preserving the spatial disposition of ocular vascular networks is cumbersome using classical histological procedures. Here, we demonstrate that light-sheet fluorescence microscopy (LFSM) of cleared mouse eyes followed by extensive virtual dissection offers a solution to this problem. To the best of our knowledge, this is the first 3D quantification of the evolution of the hyaloid vasculature and of post-occlusive venous remodeling together with the characterization of spatial distribution of various cell populations in ocular compartments, including the vitreous. These techniques will prove interesting to obtain other insights in scientific questions addressing organ-wide cell interactions.

[1] Clinical Investigation Center 1423, Quinze-Vingts hospital, INSERM-DHOS, 28 rue de Charenton, Paris F-75012, France. [2] Sorbonne Université, INSERM, CNRS, Institut de la Vision, 17 rue Moreau, F-75012 Paris, France. [3] Department of Ophthalmology, Fondation Ophtalmologique Adolphe De Rothschild, F-75019 Paris, France. [4] Department of Ophthalmology, The University of Pittsburgh School of Medicine, Pittsburgh, PA 15213, USA. [5] Univ Paris Est Créteil, INSERM, IMRB, F-94010 Créteil, France. [6] AP-HP, Groupe hospitalo-universitaire Chenevier Mondor, Centre d'investigation clinique Biothérapie, F-94010 Créteil, France. [7] These authors contributed equally: Anna Verschueren, Morgane Belle. ✉email: mpaques@15-20.fr

The choroidal, retinal and hyaloid vessels jointly support the development and metabolism of the eye, hence a comprehensive approach of ocular development, physiology and pathology requires taking into account the spatial relationship between these layers. The ocular vascular development is a complex, finely orchestrated sequence of events, involving spatial and temporal coordination of hyaloid, choroidal and retinal vessels. The hyaloid vascular system (HVS) is a developmental network which precedes the development of retinal vessels, and disappears when retinal vessels develops[1–3]. In mice, the earliest evidence of a hyaloid vasculature can be detected at embryonic day 10.5 (E10.5)[4]. From a single arterial source in the optic nerve the HVS emerges and diverges into three subnetworks: the vasa hyaloidea propria (VHP) in the vitreous, the tunica vasculosa lentis (TVL) over the posterior surface of the lens and the pupillary membrane (PM) on the anterior surface of the lens. The HVS capillaries are made of a continuous layer of endothelial cells embedded in a discontinuous layer of pericytes cells, surrounded by the basal lamina. The VHP, TVL and PM interconnect at the interface of anterior and posterior segments, and their venous drainage is thought to occur through the limbus and/or the vortex veins, although this has not been clearly determined. Around birth, the HVS regresses in parallel to the development of retinal vessels and secondary vitreous. The vitreous develops in a two stages process, a primary, cell-rich vitreous developing prior to HVS, and a secondary acellular vitreous external to the HVS[5]. The timely regression of the HVS accompanies the establishment of the adult trilayered retinal microvasculature. Metabolic support of the prenatal eye is also provided by the choriocapillaris, which develops earlier than retinal vessels[3,6].

The knowledge on the temporal and spatial coordination of the development, physiology and diseases of the ocular networks would therefore greatly benefit from a nondestructive, holistic approach by microscopy. Other areas of research on vascular diseases may also gain from a preservation of the vascular topology integrity. In retinal vein occlusion (RVO) a branch of the central vein of the retina becomes occluded, which is a common cause of visual loss in humans[7]. The venous occlusion leads to the widespread development of collateral veins partially compensating flow impairment in the affected area. Despite such collateral vein development, extensive microvascular damage may occur such as capillary dropout, leading to ischemia, chronic retinal edema and secondary neovascularization. This pathology benefits from a well-characterized rodent model[8,9]. While this disease has been studied thoroughly in the clinics[10] histological studies are few. Technical limitations of current microscopy techniques, in particular the small field of view, limits the documentation of the widespread post-RVO microvascular remodeling.

Histology of the entire eye is notoriously difficult for several reasons, including the fact that the eye encapsulates tissues of heterogeneous compositions, optical properties and firmness: these comprise the highly hydrated vitreous, the cellular-dominated neuroepithelium as well as dense fibrillar structures such as the lens, sclera and optic nerve sheaths. The vitreous is a highly hydrated gel that collapses upon sectioning; neither retinal flat-mounts nor en bloc dissection[11] can satisfactorily maintain its spatial organization. In vivo approaches have been proposed, in particular using three-dimensional OCT[12] or intravital microscopy[13] but these approaches are restricted to a narrow field of view, and do not allow cell characterization. Our understanding of the development, function and diseases of developing and adult ocular circulation would therefore be significantly improved by technologies preserving their spatial integrity.

Light-sheet fluorescence microscopy (LSFM) of cleared tissues offers such promises, due to its unique capability for three-dimensional (3D) observation of centimetric-sized samples[14–19] hence potentially better preserving the spatial configuration of ocular vessels. Recently, thanks to refinements of the solvent clearing technique iDISCO+[14], we achieved LSFM of intact albino mice eyes. Subsequently, *in toto* eye examinations of cleared eyes from various mouse strains have been reported[20–23]. Here, we analyzed developmental and pathological processes in mouse ocular vasculature using clearing, immunolabeling and LFSM. Virtual dissection and three-dimensional tracing in intact eyes allowed quantitative insights into the temporal and spatial relationship of the different ocular vascular networks.

## Results

### Eye clearing and LSFM microscopy allow comprehensive and capillary-level detailed observation of ocular vascular networks.

*In toto* eyes were processed for the analysis by LFSM using the following protocol (supplementary Fig. 1): eyes were fixed, bleached for depigmentation, stained using immunohistochemistry then cleared and imaged using the LSFM (supplementary Fig. 1a, c). Images were segmented (Supplementary Fig. 1b) and reconstructed in 3D (Supplementary Fig. 1d, e). This protocol allowed us to virtually reconstruct the ocular vasculature of intact, adult mouse eyes. A comprehensive staining of the entire ocular vasculature by anti-CD31 (Platelet endothelial cell adhesion molecule) antibody is shown (Fig. 1).

Details of the scleral, limbal, choroidal, retinal and anterior segments vessels are shown in panels Fig. 1b-d. *In toto* viewing facilitated the exploration of vessels bridging the interface of the anterior and posterior segments, such as the connections of limbal arteries to the iris (yellow arrowheads in Fig. 1a, d) and their venous drainage into the choroid. This enabled the visualization of the entire vascular pathway of posterior limbal arteries from the optic nerve to the iris (Fig. 1f) and the drainage of ciliary processes into vortex veins (Fig. 1e)[24]. The depth of view enabled the resolution of the different microvascular layers of the retina and the choroid in a given zone, showing successively the artery-dominated superficial vascular layer, capillary dominated intermediary layer, the vein-dominated deep vascular layer[25] (Fig. 1g yellow square, detailed in h) and the choroid, with its connections to vortex veins (asterisks in Fig. 1g). Virtual dissection of vascular layers showed details of each layer down to the capillary level (Fig. 1h). Other labelings (anti colIV and anti meca32) and details on the limbus, retina and optic nerve can be found in Supplementary Fig. 2, and a 3D view of the entire eye can be observed in more detail in the supplementary video 1 (https://histo.pariseyeimaging.com/Resources/5cebbd002b-Three-dimensional-characterization-of-developing-and-adult-ocular-vasculature-in-mice.en.htm).

### 3D observation of vascular development.

The concentric disposition of the retinal and HVS vasculature was preserved with minimal to no deformation throughout the different time-points. The hyaloid vasculature is challenging to document because of their loose structural support within the vitreous gel. Clearing and LSFM imaging in developing eyes enabled *in toto* observation of hyaloid and retinal vessels in their native 3D configuration. Eyes at different time points of development (from E12.5 up to P60; see also the video) were stained by anti-CD31 and anti-NG2 antibodies, staining respectively endothelial cells and pericytes then processed for iDISCO+ clearing. Figure 2 shows anti-CD31-labeled developing eyes at selected time-points. More details and 3D views are provided in the supplementary video 1 (from the beginning to 2:42). At E12.5 the paths of the two long ciliary arteries could be identified (see video at 1:03). The developing HVS (Fig. 2a) was seen in the primary vitreous; it was not

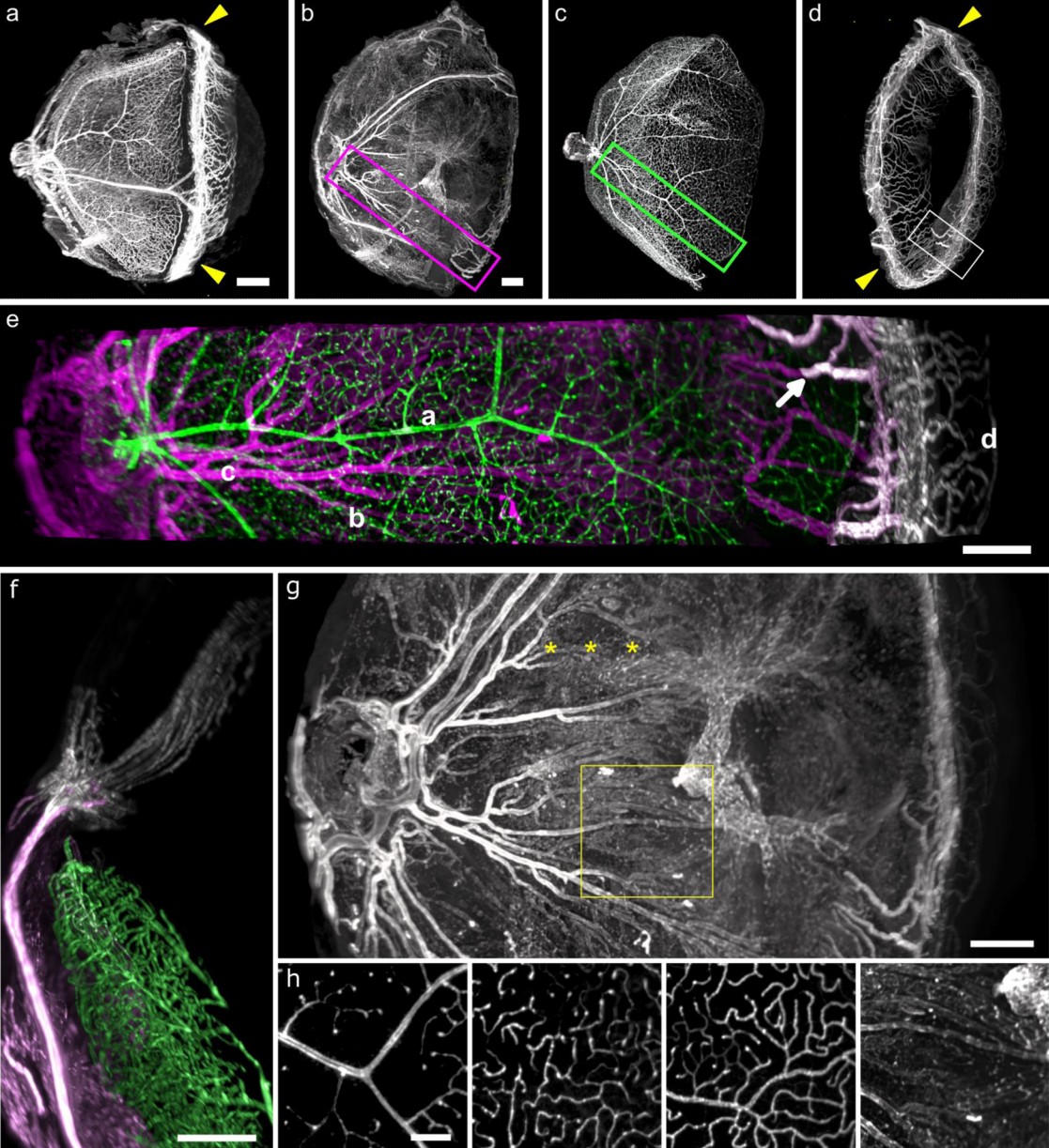

**Fig. 1 LSFM imaging of anti-CD31 immunostained cleared balb/c adult mouse ocular vasculature. a–d** *In toto* viewing showing the superimposed scleral and choroidal (**b**), retinal (**c**) and anterior segment (**d**) circulations (In **b** to **d**, the orientation of the eye was slightly rotated compared to **a**). Yellow arrowheads: limbus. **e**: superimposition of the choroid (magenta), retinal (green) and anterior segment circulation (white) from the boxed areas in **b** and **c** (**a**: retinal artery, **b**: retinal vein, **c**: ciliary arteries, **d**: iris). The arrow points to a connection between a vein of the anterior segment to the choroidal circulation. **f**: connection of a long ciliary artery (magenta) with the anterior segment circulation (white). **g**: close-up of the choroid showing ciliary arteries and a vortex vein (yellow asterisks). **h**: Details of microvascular layers in the boxed area in **g**. From left to right: superficial, intermediary and deep retinal microvascular layers, and choroid (all scale bars are 200 µm).

possible at this stage to distinguish the different subnetworks. At E14.5, the 3D organization of the vasculature had reached a higher level of complexity; the choroid is at that time a densely vascularized layer and the VHP, the TVL and the PM are visible. The long ciliary arteries resolve in a dense vascular network at the limbus. The presence of choroidal arteries and vortex veins is an indication of a functional choroidal network, suggesting that the choroidal vasculature already provides metabolic support to the developing retina.

Postnatal changes of the ocular vasculature are described in Fig. 2b, c (the choroid and sclera were numerically dissected to enable a better viewing of the underlying networks). Between P0 and P18 (from 1:40 to 3:07 in the video) the regression of hyaloid

vessels occurs concomitantly to the progression of retinal vessels. At P0, the HVS is densely vascularized, the VHP running in close contact with the retina while the TVL and PM bracket the lens. The highest microvascular density was observed at the convergence of the VHP, TVL and PM, that is, at the interface of the anterior and posterior segment, particularly around the lens equator and the ciliary bodies (Fig. 2a). At P6 (see also Supplementary Fig. 3), the vascular density of the TVL has regressed; the development of the secondary vitreous has moved the VHP away from the retina, hence likely contributing to decreased metabolic exchanges between the retina and VHP. Some capillaries from the peripheral VHP could be observed in close contact with the retina (Supplementary Fig. 4). These vessels

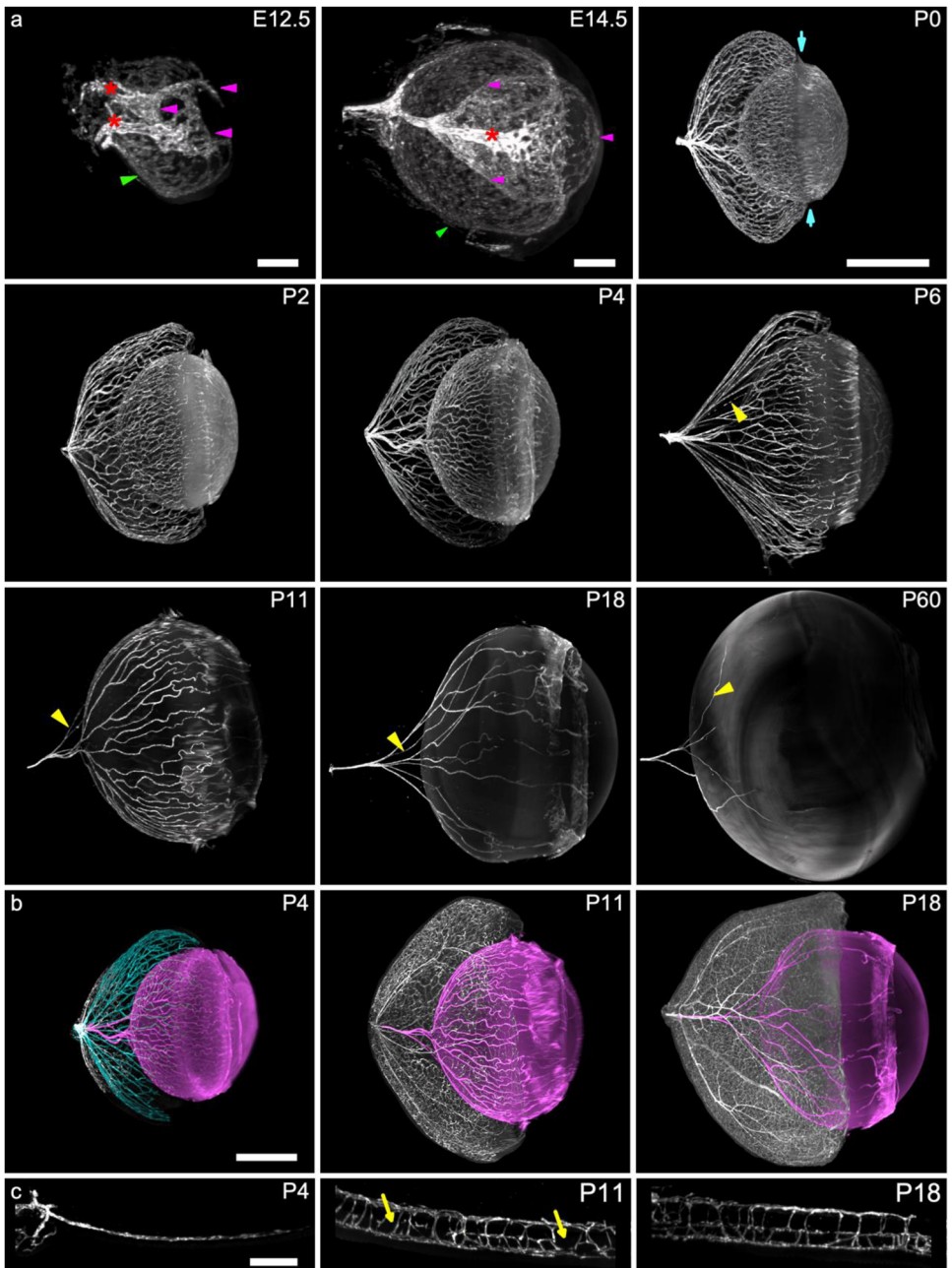

**Fig. 2 LSFM imaging of the ocular vasculature during development in the BalbC mouse (anti-CD31 immunolabeling). a** successive time-points of hyaloid vessels. Embryo eyes are shown with the choroid (green arrow heads: choroid, magenta arrow heads: HVS); At E12.5 (embryonic day 12.5) there are two long ciliary retinal arteries (red asterisk). At E14.5 the choroid (green arrow heads) is densely vascularized and the entire hyaloid system (magenta arrowheads) is clearly organized. In the subsequent time points, the choroid and retina have been numerically removed. The highest microvascular density was observed at the convergence of the VHP, TVL and PM (blue arrows in P0 for instance). **b** Superimposed hyaloid and retinal vasculatures of postnatal eyes (VHP in cyan; hyaloid arteries: TVL and PM in magenta). **c** developing retinal vasculature showing the monolayered (P0), partially (P11; asterisks indicate discontinuities of intermediate plexus) and fully trilayered (P18) structures (yellow arrowhead: regressing vessels) (Scale bars: panel **a** and **b** 500 μm except for E12.5 and E14.5 (200 μm); panel **c**: 100 μm; VHP vasa hyaloidea propria, TVL tunica vasculosa lentis, PM pupillary membrane).

connecting the hyaloid system to the retina formed a loop, none of which were found to be connected to the retinal circulation. The low level of anti-CD31 labeling of these adhering capillaries suggests that they are regressing vessels.

Figure 2 panels b and c illustrate the synchrony between the development of the retina and the regression of hyaloid vessels. At P4 the retinal vasculature shows a single superficial layer (Fig. 2b, c), while the HVS is densely vascularized. At P11, the situation is inverted, with the VHP having already regressed while

the retinal vasculature comprises two layers (Fig. 2c). At P18, the trilayered retinal vasculature is fully developed and the HVS is simplified, with only few vessels of the TVL left. In the adult, remnants of vessels were found around the lens (see supplementary video 1 for additional 3D view of the panels).

**Quantification of hyaloid networks**. We sought to quantify the evolution of the vascular topology and the vascular density of the

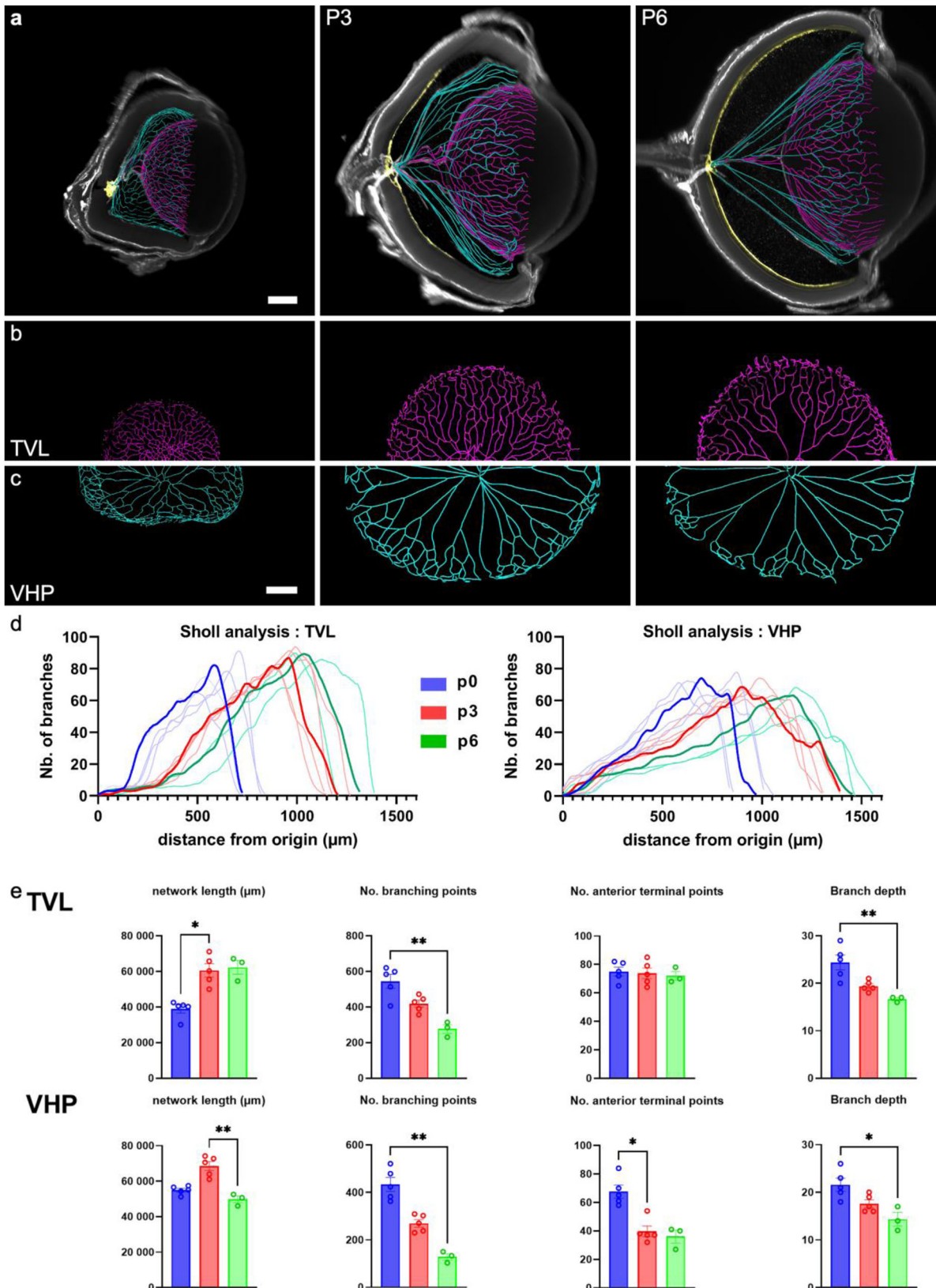

retina and hyaloid vessels during development. Previously used to quantify characteristics of dendritic processes of neurons, Sholl analysis has then been used to compare the 3D evolution of vascular networks in the eye. Its principle is based on counting the number of branches that intersect concentric spheres of increasing radii around the beginning point of the constructed

network, here defined as the optic nerve. We conducted Sholl analysis of the HVP and TVL (Fig. 3a–c) at P0, P3 and P6 (Fig. 3d; from 4:14 to 4:28 in the video). The maximum number of branches remained rather constant from P0 to P6 (i.e. around 90 for TVL and 60 for VHP). Between P0 and P6 the overall length of VHP and TVL vessels increased, in parallel to the

**Fig. 3 Morphometric analysis of the postnatal regression of hyaloid vessels. a–c** Vascular skeletons from the TVL (magenta) and VHP (cyan) superimposed with a single cross-section of the corresponding eye (the retina is highlighted in yellow). **b** and **c** show en face view of the networks (Scale bars: 200 μm; Scale bars in each line are identical). **d** Sholl analysis of VHP and TVL. Sholl analysis counts the number of branches as a function of distance from the optic nerve (distances expressed in μm); Each curve represents one eye ($n > 3$ at each time point), with a representative curve highlighted. Note the progressive stretching and corresponding shift of the shape of the curves, for both the TVL and the VHP. **e** Comparison across ages of total network length, number of branching points (number of bifurcations in the network), number of anterior terminal points (see text for details). Bar plots represent mean $+/-$ SEM. Significance were tested using a non-parametric Kruskall-Wallis test corrected using a Dunn's correction for multiple comparisons (VHP vasa hyaloidea propria, TVL tunica vasculosa lentis; *p*-values: *$p \leq 0.05$, **$p \leq 0.01$).

growth of the eye. At the same time the vascular density of the posterior VHP and TVL and the number of branchings decreased (Fig. 3e). These findings suggest that the elongation of hyaloid vessels is accompanied by simplification of the networks. Magnification of LFSM scans identifies regressing capillaries (Fig. 4b; see further description below). Hyaloid vessels at the interface of the anterior and posterior segment are the last to show capillary regression. Hence, in the postnatal posterior segment of the hyaloid complex, vascular regression predominates over angiogenesis.

**Perivascular cells during vascular development (Fig. 4).** During developmental angiogenesis, pericytes and glial cells are recruited for vessel stabilization[26,27]. As opposed to the retina, hyaloid vessels lack astrocytic coverage[2,8]. To study the pericyte coverage on the HVS and retinal vessels we labeled P0 and P6 mouse eyes using anti-NG2 antibodies. At P0, anti-NG2 stained the whole eye (Fig. 4a; from 4:29 to 4:57 in the video). Pericyte coverage could be observed on all layers of the hyaloid vasculature (Fig. 4c) and along the developing retina (Fig. 4b, top) from their onset in the optic nerve. A differential NG2 expression could be observed, with a very strong expression in the PM at P0 (Fig. 4a). This expression is still strong in the PM at P6 (Fig. 4b) but also extends to the pericytes covering the periphery of the VHP. During pruning of the hyaloid vasculature, discontinuous NG2 labeling gave capillary remnants a pearl string-like appearance (Fig. 4b, insert shows an example of a remnant capillary with a pearl string-like appearance and no CD31 labeling) that persisted longer than the CD31 immunolabeling as previously reported[28].

LSFM and clearing also allowed us to study individual cells during development (Fig. 4c). Individual Lyve-1+ cells (presumably macrophages[29]) could be observed through the entire eye, either in the vitreous, along the inner limiting membrane of the retina or closely associated with the HVS[30,31]. A decreased density of Lyve-1+ cells between P2 and P8 was observed.

The maturation of astrocytes is intrinsically linked to the maturation of the retinal vascular bed[31–33]. This is reflected by glial fibrillary acidic protein (GFAP) expression of individual astrocytes whose intensity increases in parallel to their invasion of the retinal vascular network. At P8 astrocytes near the optic nerve showed a strong expression of GFAP in a characteristic star-shape pattern (Fig. 4f), whereas peripheral astrocytes covering the immature vessels had a honeycomb-like pattern and expressed a lower level of GFAP immunoexpression (Fig. 4e).

**Vascular remodeling following venous occlusion.** To study vascular remodeling in an adult eye, we performed branch retinal vascular occlusion (BRVO) in mice by laser photocoagulation, which leads in the following weeks to vascular remodeling characterized in particular by the extensive development of tortuous vessels called collateral veins which are distinct from new vessels[8,9]. Figure 5 and the supplementary video 1 (from 4:58) show a BRVO examined both in vivo and by LFSM. In panel a, the retina is visualized in vivo using scanning laser

ophthalmoscopy fluorescein angiography. Panel b shows the LFSM of the stained retina, showing the same vessel and occlusion site. The collateral vein can be seen connecting the occluded vein to an adjacent vein, crossing under an artery demonstrating that collaterals emerge from capillaries bridging two arterial perfusion territories[9]. The supplementary video 1 shows another BRVO case.

## Discussion

Here, we showed that LSFM was able to comprehensively document the vasculature of intact eyes to the capillary level. LFSM is gaining popularity among researchers interested in the eye because of its capability to overcome many limitations of conventional histology techniques. For instance, the absence of physical sectioning suppresses the risk of inadvertent damage to regions of interest during flat-mounting. Henning et al. published the first images of LFSM of mouse vessels after clearing, using a slightly different protocol called EyeCi, which stems from the iDISCO+ procedure that we reported[14]. Yet, Henning et al. did not provide any quantitative data and no developmental or pathological data. Subsequently, several reports added their contribution to the understanding of the contribution of LSFM to ocular histology. Vigouroux et al. (2020) described developmental abnormalities related to the DCC gene defect, using a similar approach to ours (yet not quantifying the microvasculature). Yang et al. (2020) provided essentially data on the external vascularisation of the anterior segment of the eye. Ye et al. (2020) provided essentially a proof of concept of the feasibility of LFSM in pigmented eyes, but did not provide in-depth description of their observations, hence bringing little contribution to the knowledge of ocular physiopathology. Gurdita et al. (2021) provided morphometric data of the retinal and choroidal volume, and spatially characterized grafted photoreceptors. Again, no details of the organization of the microcirculation were provided. Prahst et al. (2020) provided a detailed comparison of confocal microscopy and LFSM. However, they provide no data of the eye in toto; indeed, all retinal samples were dissected from the choroid and sclera. By comparison with the abovementioned papers, we provide in our paper a more detailed, quantitative description of the different vascular layers without sectioning together with extensive numerical dissection, showing in particular the connections between anterior and posterior segments circulations, and a comprehensive, quantitative, longitudinal overview of the hyaloid vasculature, showing that the interface of anterior and posterior segments maintained a high capillary density late in development. We also showed that LFSM contributed to identify several features of the developing vascular anatomy, such as connections between hyaloid vessels and the peripheral retina.

LFSM enabled fine spatial and temporal investigations of the developmental and regression cycles of vascular layers, and a better understanding of their anatomical relationship. Avoiding sectioning also allowed the preservation of the vascular connections between anterior and posterior segments and the disposition of vessels in the vitreous. In adult as well as in developing eyes,

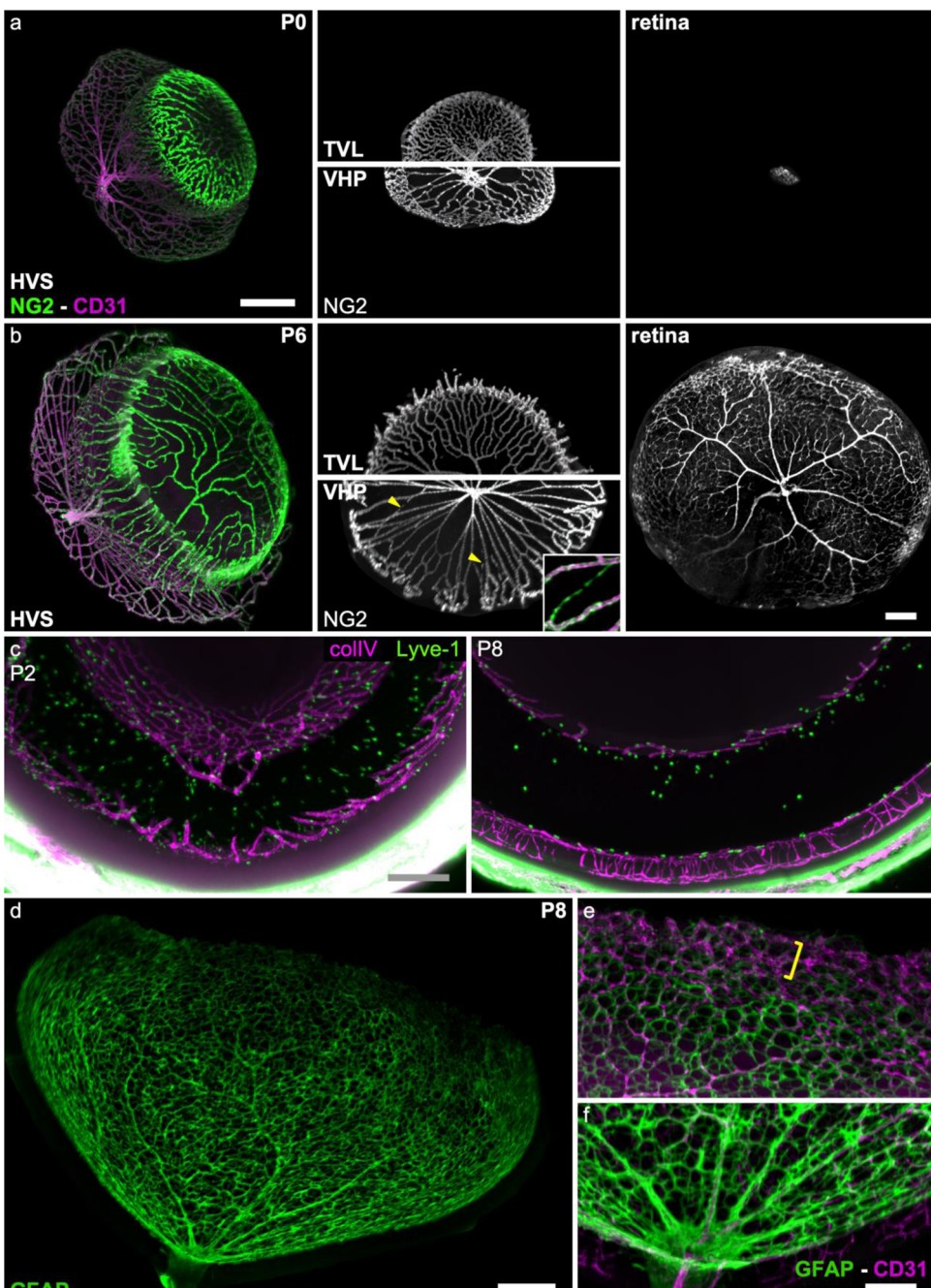

**Fig. 4 perivascular cells during postnatal development. a**, **b** NG2 (pericytes, green) and CD31 (endothelial cells, magenta) expression in P0 and P6 hyaloid and retinal vessels. Note that NG2-expressing cells are present at the earliest stage of hyaloid and retinal vascular development in all layers. First column shows a 3D view of all hyaloid vasculatures, second and third show an en-face view of TVL, VHP and retina NG2 expression, respectively. Yellow arrowheads: regressing vessels, insert: detail of a regressing vessel with double staining: anti-NG2 green, anti-CD31 magenta showing the pearl string-like appearance, where only the endothelial cell remains. **c** Lyve-1-expressing hyalocytes' presence in the vitreous and around the HVS. A portion of those hyalocytes surround the hyaloid blood vessels. Between P2 and P8 note the drastic decrease in the number of observed hyalocytes in the vitreous. **d**–**f** Astrocytic pattern in the developing retina (P8) through GFAP expression. LFSM of anti-GFAP (green) and anti-CD31 (magenta) immunolabeled retina at P6 showing the glial scaffold accompanying the development of the vasculature. The GFAP negative tip of the vascular front (not covered by astrocytes) is bracketed E. scale bar (Scale bars in each line are identical for panels **a**, **b**, **c**. Scale bars values:a, **c**, **d**: 200 µm; **b** and **e**: 100 µm; VHP:vasa hyaloidea propria, TVL tunica vasculosa lentis).

the highest capillary density was observed at the interface of the anterior and posterior segments at the equator of the lens. Indeed, the HVP, TVL and PM converged to this area, which also bears the higher capillary density in adult eyes. Assuming that capillary density reflects metabolic activity, this suggests that this area, which contains the lens, is the most metabolically active of the

eye. However, as iDISCO+ process is known to change volume up to 10%, those results are to be taken comparatively, and not as absolute values. Examining intact eyes also allowed us to characterize poorly documented connections such as between the avascular, developing retinal periphery and the VHP and in adults between anterior and posterior segments, showing for

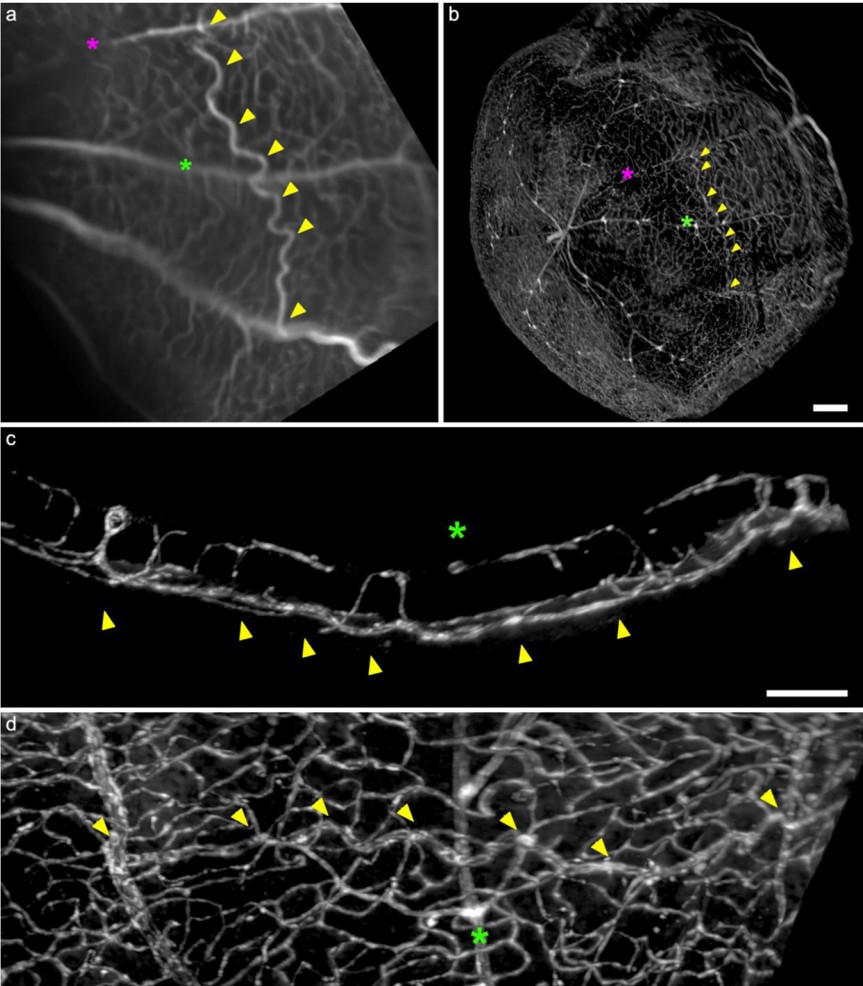

**Fig. 5 In vivo imaging and LFSM of an anti-CD31 immunostaining mouse eye after experimental retinal vein occlusion. a** in vivo scanning laser ophthalmoscopy of the collateral vessel (arrowheads) running under an artery (green asterisk); the site of the occlusion is shown by the magenta asterisk, clearly visible by the lack of fluorescein perfusion; the collateral vessel is easily differentiated from other vessels by in-vivo ophthalmoscopy, characterized by a tortuous phenotype. **b** in toto LFSM of the same eye after anti-CD31 immunolabeling. The collateral vessel can be found using its 3D correspondence to in-vivo imaging. **c** and **d**, close-ups showing that the collateral vessel (arrowheads) is located in the deep vascular layer, under the artery (green asterisk) (Scale Bars: **a** and **b:** 200 μm; **c** and **d**, 50 μm).

instance the drainage of the anterior segment through the vortex veins as we previously reported in vivo[24].

Using the capacities of LSFM and clearing of going from organ-wide imaging to cell scale analysis, we were able to study perivascular cells surrounding the regressing HVS. NG2 + cells were shown surrounding all layers of the VHS until regression, their pearl-like appearance along regressing vessels preserved in the nondissected eye. Vitreal Lyve-1+ cells could also be precisely localized through the entire eye during development.

Microvascular remodeling in BRVO models has also proven complex to document using classical histology. To the best of our knowledge, we provide here the first spatially-preserved reconstruction of the pattern of collateral vein development following BRVO. Figure 5 shows that establishing a correspondence between the in vivo imaging by SLO and histology by LFSM was possible and ensured proper identification of a collateral vessel that may otherwise be difficult to detect within the dense mesh of retinal vessels post-mortem. The dilated collateral channels could be followed in 3D through the entire retina and all vascular layers and studied from all angles, while also imaging the occlusion site and the entire vascular environment around the post-occlusive remodeling. This enabled it to detect affected veins in distally located areas, in particular in the extreme periphery of the retina. Indeed, a single-branch retinal vein occlusion affects roughly half of the vasculature because of the deviation of venous flow, which finds its way out through adjacent channels (which also runs along the extreme periphery of the retina). Even more, it shows that LFSM enables better documentation of the extent of venous dilation than in vivo imaging. It also demonstrates that the vascular dilation and tortuosity is preserved to some extent during tissue processing.

A specific drawback of LFSM in the intact eye is the refraction of incident light caused by the lens, which may blur the side of the sample opposite to incident light; this may necessitate rescan of the sample with a different orientation. The short duration of the acquisitions mitigates this inconvenience (less than 15 min to acquire an entire mouse eye). The short time of acquisition by LSFM also allowed us to process and image many samples at the same time, facilitating access to a complete timeline of the ocular vascular development from embryo to adult. s. Post-processing of LFSM data can participate in reducing the need for repeated animal experiments.

Therefore, LFSM of cleared intact mouse eyes opens avenues for a global, quantitative, three-dimensional exploration of ocular

development and diseases. It indeed allows more precise modeling of the circulation of developing and adult eyes, and the detailed comparison of vascular architecture over time and space. We believe that implementing clearing and LSFM in ocular research will permit a switch from a retina-centric view of the development and pathology to a more global, organ-wide, interconnected approach of the eye as a whole. This approach will offer a powerful tool for anatomical studies, pharmaceutical screening, or mutant characterization.

## Methods

**Animals**. The protocol, comprising the scientific goal and experimental procedures, was approved by the Sorbonne Université ethic committee (APAFiS #19703). BALB/c adult and embryonic mice were purchased from Janvier Labs (St-Ile Le Genest, France). Euthanasia procedures were as follows: pups between P0 and P14 were euthanized by decapitation; P14 and older (including pregnant females) were euthanized by carbon dioxide inhalation. Post-natal eyes were harvested, prefixated in 4% PFA for 30 min, periocular tissues were removed and eyeballs were fixated in 4% PFA overnight. Embryos were fixated whole in 4% PFA overnight. Males and females were used indifferently. As a rule, experiments were done at least in triplicate.

**Retinal vein occlusion**. Branch retinal vein occlusion was performed by laser photocoagulation using a previously reported procedure[9]. Briefly, after pupil dilation with tropicamide, NO2 anesthetized mice were placed in front of a slit lamp. Fluorescein sodium (Novartis) was administered intraperitoneally (50 μl of a 1 mg/ml solution) to visualize vessels and to potentiate the thermal effect of laser. The fundus was visualized through a non-contact SuperPupil lens (Volk, Mentor, Ohio). Care was taken to obtain accurate focusing of the aiming beam on the target vessel. A thermal laser shot (wavelength 534 nm; Quantel medical, Gournon d'Auvergne, France) with the following parameters: duration 0.5 s; power 100-150 mW; spot size 50 μm) was placed on a superior retinal vein, approximately 2 disc's diameters from the disc. Two to five impacts were usually sufficient to obtain complete occlusion. In the following days, the presence of collateral vessels was checked by scanning laser ophthalmoscopy fluorescein angiography (Heidelberg Engineering, Heidelberg, Germany). Animals were sacrificed at day 30 and their eyes processed for LFSM microscopy.

**Immunostaining**. Immunostaining protocols were inspired from[16]. Several steps were customized to the specificities of the developing and adult mouse eyes. The first step consists in depigmentation. Briefly, fixated eyes were first dehydrated in PBS/methanol baths (PBS 10X, Gibco 14200-083, Invitrogen; methanol, VWR - 20847.360), scaling up 50%, 80% to 100% methanol (1h30/bath). The dehydrated eyes were then bleached overnight in methanol and 6% $H_2O_2$ (VWR 23613.297) overnight at +4 °C. Our protocol is here optimized for albino mouse eyes, and thus make use of a mitigated bleaching. The next day, eyes were rehydrated in gradual PBS/methanol baths (1h30, 100%, 80%, 50% methanol then PBS baths). The eyes were then permeabilized in PBSGT-0.5X (1X PBS, 0.2% Gelatin (PROLABO 24350.262 VWR), 0.5%Triton X100 (SIGMA- X100)) for 24 h to 48 h, +4 °C, gentle stirring. All stirring and incubations were perfomed in the Incu-shaker mini from Benchmark scientific. Primary antibodies were diluted in PBSGT-0,5X and 10 μg/mL saponin (SIGMA-S4521). For this paper, we used primary antibodies against COLLIV (Biorad 134001), CD31 (PECAM-1 BD bioscience 550274), NG2 (MERCK-Millipore ab5320), GFAP (DAKO Z0334), alpha-SMA (Sigma-Aldrich MERCK-Millipore A2547), meca32 (BD pharmingen 550563) and Lyve-1 (Biotechne, AF2125). The following dilutions were used: 1/600 for anti-CD31, anti-colIV, anti-NG2 and anti-GFAP, 1/500 for anti-meca32, and 1/100 for anti-Lyve1 antibody. Eyes were incubated in primary antibodies between 7 to 10 days (7 days for the smaller samples: early developmental stages like P2 eyes, and 10 days for adult eyes), 37 °C and 70 RPM for a 1/5. After primary labeling, eyes were washed in PBSGT- 0.5X (x3). Secondary antibodies (Cross-Adsorbed Secondary Antibody from Thermofisher, precisely Alexa 546, 647 and 790 nm) were diluted in PBSGT-0.5X and 10 μg/mL saponin at a 1/500 dilution, with TO-PRO™-3 Iodide (T3605, Thermofisher scientific, dilution 1/100) as a nuclear staining and filtered using a 0,22 μm syringe filter (ROTH- KH54.1). The eyes were incubated up to 24 hours, 37 °C, 70 RPM before being washed in PBS (x3).

**Clearing**. The labeled eyes were first embedded in agarose (1,5X, in TAE; ROTH-2267.4), to facilitate processing. We used methanol clearing from the iDISCO+ protocol (Renier 2016) to preserve the size and sphericity of the eye samples.The samples were then dehydrated by successive PBS/methanol baths (1 h/each, 20%, 40%, 60%, 80% 100%) under a fume hood by agitation at 12 rpm (SB3 tube rotator, Stuart) at room temperature in 15 ml TPP tubes. Once dehydration completed, samples were incubated ON in 2/3 Dichloromethane (SIGMA-270997) and 1/3 methanol. The next morning, samples were incubated 30 min in 100% dichloromethane before filling the tube with 100% Dibenzylether (DBE; SIGMA-108014).

After waiting at least 24 hours, samples were considered cleared and ready for imaging.

**Imaging**. Acquisitions were performed by using an ultramicroscope I with the Inspector Pro software 5.0.285.1 (LaVision BioTec) and a binocular stereo-microscope (MXV10, Olympus) with a 2X objective (MV PLAPO, Olympus) at different magnifications (from 2X to 6.3X) (See Supplementary Table 1 for image-by-image details). The light sheet was generated by an OBIS laser diode (wavelength 561 or 635 and 785 nm, Coherent Sapphire Laser, LaVision BioTec) and focused using two cylindrical lenses. Two adjustable protective lenses were applied for small and large working distances. A binocular stereomicroscope (MXV10, Olympus) with a 2x objective (MVPLAPO, Olympus) was used at different magnifications (2x, 2.5x, 3.2x and 6.3x) (see supplementary table 1 for corresponding zoom factors and numerical apertures).

Samples were placed in the microscope's quartz cuvette filled with DBE mounted on a motorized XYZ-stage for moving imaging chamber and sample holder. The step between each image was fixed at 1 and 2 μm (See Supplementary Table 1 for image-by-image details). All tiff images are generated in 16-bit. For large samples, a platform was created using PDMS (Sylgard) fixed to a sample holder. The total acquisition time for the samples ranged from 15 min to 1 h, depending on the number of wavelengths acquired, the total z step and the magnification needed. For 3D imaging using the confocal microscope (Olympus, FV1000), homemade glass chambers containing DBE (keeping the sample clear) were used for the imaging.

**Image processing and analysis**. After acquisition, images sequences were imported into Imaris software (versions 9.5 to 9.7, Bitplane) to be studied in 2D or 3D. In order to elucidate the complexity of the 3D eye, numerous tools from the software were used: Slicers and orthoslicers to isolate parts of the samples; the surface tool to delimitate and isolate various ocular structures and vascular beds; the filament plugin to trace and analyze the vascular networks of hyaloid vessels, using 'edit manually' settings, and the 'autopath (no loop)' algorithm and 'calculate diameter of filament form image' options. Inbuilt Imaris' animation and snapshot tools were used for video and image acquisitions. We use Fiji (Schindelin et al., 2012) and FIJI's plugin ScientiFig[34,35] for scientific figures creation, and iMovie© (10.1.11) for video editing and legends generation.

**Statistics and reproducibility**. When comparing the different networks parameters, we were interested in observing the total network length, the total number of branching point (the number of bifurcations in the network), the number of anterior terminal points (define as the number of terminations of the network at the anterior end), and the maximal branch depth. The latter was defined as the number of bifurcations in the shortest path from the beginning point at the optic nerve to a given point in the network. The maximal branch depth is the highest value of branching depth for the considered network. Each time point was replicated in more than three independent eyes (from different animals): for P0 $n = 5$, for P3 $n = 5$, for P6 $n = 3$. Sholl analysis curves were smoothed using a 50 μm wide window. All statistics were performed using GraphPad (Prism 8). We used mean and standard error mean calculation for comparison. Significance was tested using a non-parametric Kruskall-Wallis test corrected using a Dunn's correction for multiple comparisons.

**Reporting summary**. Further information on research design is available in the Nature Research Reporting Summary linked to this article.

## Data availability

Data available at: https://figshare.com/projects/Three_dimensional_characterization_of_developing_and_adult_ocular_vasculature_in_mice_using_in_toto_clearing_/143619

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

## Acknowledgements
This project was funded by the Region Ile-De-France (EX047007 - SESAME 2019 - 4DEye), LabEx LIFESENSES (ANR-10-LABX-65), Institut Hospitalo-Universitaire FOReSIGHT (ANR-18-IAHU-01) and ANR Normather (ANR-14-CE16-0023). The funding organizations had no role in the design or conduct of this research.

## Author contributions
M.D., M.B., I.C., and M.P. initiated and conceived the project. M.D., A.C., M.P. Designed the project. M.B. and S.F. contributed essential technical expertise. M.D., M.B. and M.P. performed research. M.D. generated the results. M.D., A.V., M.B., L.B. collected data. M.D., A.V. Analyzed the data. A.V. Designed and performed statistical analysis. M.D., A.V. and M.P. wrote the paper. A.C. and I.C. revised the manuscript. J.A.S. and M.P. Supervised the research.

## Competing interests
The authors declare no competing interests.
