## [Peer Review File · Communications Biology]

Reviewers' comments:

Reviewer #1 (Remarks to the Author):

Marie et al. present a thorough study of vascular development in murine eyes. This is a well written manuscript with very nice 3D imaging of intact eyes.

I found the paper enjoyable to read and commend the authors on the significant amount of work. My questions mainly center on the technical aspects, as this is where my expertise lies.

I have organized my comments into "major" and "minor", with suggested action items for each.

Major:

1. The lack of technical details for the imaging and image analysis pipelines make it difficult to evaluate the quantitative findings. For example, the authors state that they use step sizes of 1 to 2 mm between planes for the LSFM imaging. This is clearly not possible. Please provide a supplemental table with detailed imaging conditions for all presented data, organized by figure. I suggest the following paper as a guidelines: <https://www.nature.com/articles/s41592-021-01156-w>

2. Similarly, there is a lack of details on the image analysis. Please provide exact details on the workflows used. Preferably, code would be provided paired with representative data. For example, what is the x-axis in Figure 3D? Pixels? microns?

3. Why does Figure 3E use standard error of the mean? Is each measurement the mean of a sample group (multiple eyes)? Or is each measurement the mean of one eye? Does the data justify a non-parametric fit?

4. How efficient is the H2O2 depigmentation? It would be helpful to have a before and after image that shows the autofluorescence levels.

Minor:

1. What is the XYZ resolution of the Ultramicroscope as configured? Is the data Nyquist sampled in all three dimensions?

2. How were antibody clones and concentrations optimized?

3. Does the agar embedded eye change volume when cleared? Our experience is that the volume does slightly change.

Doug Shepherd

Reviewer #2 (Remarks to the Author):

In this manuscript the authors present the use of an organic-solvent based clearing technique to visualize the vasculature in albino mice in three dimensions using light sheet microscopy. Using this 3D-immunohistochemistry approach, they describe the ontogenetic development of hyaloid, choroidal and retinal vessels in BALB/c mice from E12.5 to P60, before demonstrating the power of the method for the characterization of retinal pathologies in a mouse model of vascular remodeling after venous occlusion.

This is a nice manuscript with powerful figures. It demonstrates the superior value of 3D histology over traditional methods and promises to promote more the widespread use of clearing technology, which ultimately could lead to valuable 3D histological databases and the reduction of animals needed for basis research. I think the manuscript is suitable for the journal but requires some revisions before it can be accepted.

Given the strong technological focus of the manuscript, the introduction falls short in acknowledging the achievements of other groups that have demonstrated the use of clearing techniques to study eye morphology. Please describe in more detail what other protocols have achieved (e.g. vasculature of the mouse eye has been described by Henning et al. as well) and which improvements the presented technique offers or which knowledge gaps are attempted to be closed. Also, please remove the poster reference (#21), it is not necessary and not good scientific practice to claim technological inventions retrospectively using references to non-peer reviewed literature.

Other comments:

Line 57: space missing after ‘‘determined’’.

Line 82-92: This paragraph needs to be deleted as it appears twice.

Line 124: Change mouse’s eye to mouse eye

Line 127: I could not find the corneal vessels in Fig 1, in which panel are they shown? Also, on what basis was the distinction between these systems made, i.e. how was the border between retinal and anterior segments defined and identified?

Line 134: Should read supplementary figure 3

Line 145: Should read supplementary figure 2

Lines 257-266: This paragraph should be deleted as it appears twice.

Lines 355-375: Some more details on the immunostaining would be helpful to increase the reproducibility:

1. Mention that the described bleaching procedure does only work for the eyes of albino animals. Vigouroux et al described a more versatile bleaching protocol for pigmented eyes that should be mentioned here.
2. Is there a specific reason why the methanol series were diluted in PBS and not in water as in the original iDISCO protocol? If e.g. this is important to minimize deformations during dehydration that should be mentioned.
3. Please give the concentrations of all antibodies used (including the secondaries and their conjugation)
4. Line 370: What kind of shaker was used for primary ab incubations?

Line 556: There are no histograms shown in this figure. An explanation of the statistical asteriscs is missing in the figure legend.

All figure legends: Please list all abbreviations used in each figure in the respective legend.

Reviewer #3 (Remarks to the Author):

Summary: The authors demonstrate, as a proof of principle, the application of tissue clearing and light sheet fluorescent microscopy (LSFM) towards the imaging and analysis of retinal and hyaloid vessel

development. The authors provide beautiful and high resolution light sheet acquired images of developing eyes and demonstrate first proof of principle utility for using this approach to study hyaloid vessel formation. The authors also demonstrate a proof of principle application for studying branch retinal vein occlusion (BRVO) models. However, there are several limitations to this manuscript, including, the already demonstrated the utility of tissue clearing and light sheet microscopy towards studying ocular vasculature from several groups (Henning et al., 2019, Vigouroux et al, 2020; Ye et al., 2020, Prahst et al 2020, Gurdita et al., 2020), so it is unclear what major advances this study provides to the field. Similarly, although a correlation between OCT and LSFM images in the BRVO model are demonstrated, it is unclear what new insights this provides.

Major Concerns:

It is unclear how the findings in this study differ, compare or add to the existing understanding of retinal vascularization and hyaloid development using traditional approaches ie. serial reconstruction or wholmount.

The study lacks a comparison to existing tissue clearing and LSFM approaches which are now well documented. Thus, the paper would be improved by using this approach to general novel insights in hyaloid development/defects and BRVO models.

Minor problems and concerns:

Line 82: Duplicated paragraph

Line 119: If albino eyes were used then why was bleaching necessary?

Line 130: Can direction of blood flow be inferred from static images? Is this based on anatomy alone? If this is common knowledge in the field can the authors clarify this to make the manuscript more accessible to a non-expert.

Line 153: Confusing to mention NG2 staining here because it is not discussed until a later section.

Line 168: What does 'numerically' dissected mean?

Line 179: Supplemental Figure 4, why is the CD31 stain so saturated?

Line 182: Supplemental Figure 5, it would be useful to have a low magnification image here to help spatially orient the reader.

Line 183: Can one draw any conclusions about function with a static IHC image in this context?

Line 192: Indicate figure.

Line 213: Is this the correct figure reference? If it's supposed to be 3B then it is also not cited in the correct order in the paragraph.

Line 226: P0 instead of P6.

Line 228: Point the discontinuous NG2 labelling more clearly in the figure. Are you referring to the green vessel only in the inset?

Line 240: The figure 4F is cited out of order.

Line 254: It's not appropriate to use 'back and forth' since this analysis has not been demonstrated to be bidirectional.

Line 257: Duplicated section.

Line 273: Should acknowledge that this is not the first attempt at imaging vessels at the capillary level and should compare to existing cleared eye literature and state the advance.

Line 315: Confocal microscopy images were not provided.

Line 369: Statements such as 'up to 10 days for primary antibody incubation' should be clarified so that readers may be able to easily reproduce this approach. At the very least, a minimum antibody incubation time should be provided. Alternatively, if incubation times differed depending on the primary antibody, then each primary antibody incubation could be specified. These primary and secondary antibody incubation times are also starkly different from existing eye clearing protocols.

Can the authors comment on how they settled on these incubation times?

Line 376: Why wasn't EyeDISCO used for the clearing procedure? It may be helpful to acknowledge differences between this paper's use of the standard iDISCO and the optimized EyeDISCO protocols.

Line 418: Why were non-parametric statistical tests used? Were parametric test assumptions violated?

Line 551: Panel D, label Y axis in figure

Line 555: Panel E, label on Y axis instead of X axis. Combine graphs on one page to use legend once.

General comment: We thank the Reviewers for their helpful comments. We took into account all of their remarks, which significantly improved the paper. Among other changes, we extended the title that now specifies that in toto eyes were examined, and suppressed a supplementary figure we felt was redundant.

Reviewer #1 (Remarks to the Author): Marie et al. present a thorough study of vascular development in murine eyes. This is a well written manuscript with very nice 3D imaging of intact eyes.

I found the paper enjoyable to read and commend the authors on the significant amount of work. My questions mainly center on the technical aspects, as this is where my expertise lies.

I have organized my comments into "major" and "minor", with suggested action items for each.

Major:

	answers :
1. The lack of technical details for the imaging and image analysis pipelines make it difficult to evaluate the quantitative findings. For example, the authors state that they use step sizes of 1 to 2 mm between planes for the LSFM imaging. This is clearly not possible. Please provide a supplemental table with detailed imaging conditions for all presented data, organized by figure. I suggest the following paper as a guidelines: https://www.nature.com/articles/s41592-021-01156-w	Using the recommended publication, we add complementary information in the method section:  - We added information on the light source, the excitation and emission wavelength bandwidth: excitation 561nm - emission 595/40nm for Alexa 546; excitation 640nm - emission 690/50nm for Alexa 647; excitation 785nm - emission 845/55nm for Alexa 790. laser intensity 100% Light source OBIS laser diode, 561 nm, 640 nm and 785 nm for respective corresponding Alexa. - Information on the exact Alexa fluorophores used was added I335. - We added the suggested supplementary table providing for each image the magnification, pixel size and numerical aperture. We also corrected the typo in the unit of the z-step : the z-step is indeed 1 to 2 μm (not 1 to 2 mm). - We precisised the motorized

	components (Motorized XYZ-stage for moving imaging chamber and sample holder.) and the version of the ultramicroscope software: Inspector pro 5.0.285.1.
2. Similarly, there is a lack of details on the image analysis. Please provide exact details on the workflows used. Preferably, code would be provided paired with representative data. For example, what is the x-axis in Figure 3D? Pixels? microns?	Image analysis was based on the filament tool of the Imaris software, that we used in manual mode. We reconstructed hyaloid vascularization using 'edit manually' settings, and the 'autopath (no loop)' algorithm and 'calculate diameter of filament form image' option. Afterwhile, we checked the accuracy of the reconstruction toward the observed image (in terms of tortuosity mainly). We added those image analysis details in the method section, I 373 Imaris software being a commercial software, the underlying code is not accessible. We apologize for the missing unit in Figure 3D, which is expressed in μm (see corrected Fig3D and legend I506).
3. Why does Figure 3E use standard error of the mean? Is each measurement the mean of a sample group (multiple eyes)? Or is each measurement the mean of one eye? Does the data justify a non-parametric fit?	Each value corresponds to one eye; Each column is the mean of 3 to 6 values. Standard error of the mean was plotted to allow comparison of the means, and was preferred to standard deviation which is more useful for comparison of the data distribution (as all data were individually plotted on the graph, standard deviation would be redundant information). Non-parametric tests were used because there is too little data to perform a normality test.
4. How efficient is the H2O2 depigmentation? It would be helpful to have a before and after image that shows the autofluorescence levels.	The H2O2 depigmentation was highly efficient in this protocol since it was only applied to albino eyes. In these tissues, there was no melanin to bleach, only hemoglobin, so no heavy bleaching protocol was needed. The depigmentation protocol is similar to Belle et al,2017. Hence, we did not provide autofluorescence images.

Minor:

	answers
--	----------------

1. What is the XYZ resolution of the Ultramicroscope as configured? Is the data Nyquist sampled in all three dimensions?	We provided the XY and Z resolutions (voxels not being isotrope in our setting) of our images, which follow the configured Ultramicroscope settings, in the supplementary table. As mentioned in the additional table, data are not sampled in XYZ resolution.
2. How were antibody clones and concentrations optimized?	We tested different concentrations for optimization, starting with our current protocol for flat-mounted retina (Darche et al, 2020; Cossuta et al, 2019) . After stepwise adjustment (“ trial and error”), we found that the optimal concentrations for light-sheet imaging were generally to be used at the same or lower (up to two fold) dilution.
3. Does the agar embedded eye change volume when cleared? Our experience is that the volume does slightly change.	As reported in Renier et al, 2016 and Belle et al, 2017, no/minimal shrinkage was observed in our experiments. Indeed, the iDISCO+ protocol was specifically developed to minimize shrinkage compared to the 3DISCO protocol. No tissue distortion was detectable in our samples; if there was any shrinkage it affected all layers equally , hence not modifying the spatial relationship between the structures which was our main concern. As proof, few retinal detachments were observed in samples (samples with retinal detachments were not processed). We added ‘As iDISCO+ process is known to change volume up to 10%, those results are to be taken comparatively, and not as an absolute network length value in the different developmental stages.’ 247

Doug Shepherd

Reviewer #2 (Remarks to the Author): In this manuscript the authors present the use of an organic-solvent based clearing technique to visualize the vasculature in albino mice in three dimensions using light sheet microscopy. Using this 3D-immunohistochemistry approach, they describe the ontogenetic development of hyaloid, choroidal and retinal vessels in BALB/c mice from E12.5 to P60, before demonstrating the power of the method

for the characterization of retinal pathologies in a mouse model of vascular remodeling after venous occlusion.

This is a nice manuscript with powerful figures. It demonstrates the superior value of 3D histology over traditional methods and promises to promote more the widespread use of clearing technology, which ultimately could lead to valuable 3D histological databases and the reduction of animals needed for basis research. I think the manuscript is suitable for the journal but requires some revisions before it can be accepted.

Given the strong technological focus of the manuscript, the introduction falls short in acknowledging the achievements of other groups that have demonstrated the use of clearing techniques to study eye morphology.

Please describe in more detail what other protocols have achieved (e.g. vasculature of the mouse eye has been described by Henning et al. as well) and which improvements the presented technique offers or which knowledge gaps are attempted to be closed.	We agree that we should have provided more elements of comparison with the achievements of other teams, which would have better highlighted the specific interest of our research. As pointed by Reviewer 2, Henning et al. published the first images of LFSM of the adult choroidal and retinal mouse vessels after clearing, using a slightly different protocol called EyeCi, which stems from the iDISCO+ procedure that we reported (Renier et al, 2016). Yet, Henning e al. did not provide any quantitative data and no developmental or pathological data. In the following years, several papers added their contribution to the understanding of the contribution of LFSM to ocular histology. -Vigouroux et al (2020) described developmental abnormalities related to the DCC gene defect, using a similar approach to ours (yet not observing nor quantifying blood vessels and microvasculature).-Yang et al. (2020) provided essentially data on the external vascularisation of the anterior segment of the eye.-Ye et al (2020). provided essentially a proof of concept of the feasibility of LFSM in pigmented eyes, but did not provide in-depth description of their observations, hence bringing little contribution to the knowledge of ocular physiopathology.-Gurdita et al. (2021) provided morphometric data of the retinal and choroidal volume, and spatially characterized grafted photoreceptors. Again, no details of the organization of the microvasculature were provided.-Prahst et al (2020). provided a detailed comparison of confocal microscopy and
---	---

	LFSM. However, they provide no data of the eye in toto; indeed, all retinal samples were dissected from the choroid and sclera. By comparison with the abovementioned papers, we provide in our paper the following: 1-a more detailed, quantitative description of the different vascular layers at the microscopic level over a large field (i.e. the eye in toto) together with fine, multilayer numerical dissection, showing in particular the connections between anterior and posterior segments circulations 2- a comprehensive, quantitative, longitudinal overview of the hyaloid vasculature, showing that the interface of anterior and posterior segments maintained a high capillary density late in development. 3-the identification of novel features such as contacts between hyaloid vessels and the peripheral retina 4-The study of vascular environnement in toto, with perivascular cells such as pericytes, astrocytes, and hyalocytes in their 3D organization through development. Their crucial role for a functional, physiological vascular network is often explored through classical histological means, but has not been described in toto in correlation to vascular development. We added this complementary information in the discussion.
Also, please remove the poster reference (#21), it is not necessary and not good scientific practice to claim technological inventions retrospectively using references to non-peer reviewed literature.	This has been corrected.

Other comments:

Line 57: space missing after "determined".	152, corrected
Line 82-92: This paragraph needs to be deleted as it appears twice.	This has been corrected
Line 124: Change mouse's eye to mouse eye.	198 corrected
Line 127: I could not find the corneal	We apologize for this error. We meant

vessels in Fig 1, in which panel are they shown? Also, on what basis was the distinction between these systems made, i.e. how was the border between retinal and anterior segments defined and identified?	“anterior segment (i.e. iris)” instead of “corneal” vessels I101. We added in figure 1A and 1D yellow arrowheads showing limbal vessels. The Reviewer mentioned the issue of the definition of the border of retinal and anterior segment. Retinal vessels are not connected to anterior segment vessels, hence providing a natural border. However, more difficult is the definition of the anatomical border between anterior and posterior segments which is conventionally defined by a line passing through the limbus and behind the lens; there are vessels that crosses this border- most notably ciliary arteries perfusing the iris (and efferent veins from the iris and ciliary bodies). For intraocular circulation, we arbitrarily defined such a border between anterior and posterior segment vessels by a virtual line passing at the posterior limit of ciliary processes; hence, the lens, iris, limbus and ciliary processes were considered as being part of the anterior segment circulation.
Line 134: Should read supplementary figure 3	This has been corrected, making sure that the supplementary figures are numbered in their order of appearance in all files.
Line 145: Should read supplementary figure 2	This has been corrected, making sure that the supplementary figures are numbered in their order of appearance in all files.
Lines 257-266: This paragraph should be deleted as it appears twice.	We deleted the paragraph.
Lines 355-375: Some more details on the immunostaining would be helpful to increase the reproducibility:  1. Mention that the described bleaching procedure does only work for the eyes of albino animals. Vigouroux et al described a more versatile bleaching protocol for pigmented eyes that should be mentioned here. 2. Is there a specific reason why the methanol series were diluted in PBS and 	 1-We mentioned in the revised paper that this protocol is not optimized for pigmented eyes. This has been added in the “immunostaining” part of the methods section (I320).Vigouroux et al describes a protocol with stronger bleaching, adapted to pigmented eyes, but unrelated to our project on albino eyes, in which we favored a more moderate bleaching. 2- This protocol is not based on the original iDISCO protocol, but the 3DISCO+ protocol published in Belle et al, 2017, which describes an optimal protocol of methanol diluted in PBS and not water. PBS has been

not in water as in the original iDISCO protocol? If e.g. this is important to minimize deformations during dehydration that should be mentioned. 3. Please give the concentrations of all antibodies used (including the secondaries and their conjugation) 4. Line 370: What kind of shaker was used for primary ab incubations?	used in their protocol to allow for a smoother dehydration with less bloating of the sample, so less deformations. 3- This has been added to the methods section. l330 4- We used the Incu-shaker mini from Benchmark scientific. This has been added to the methods section. l325
Line 556: There are no histograms shown in this figure. An explanation of the statistical asterisks is missing in the figure legend.	Fig3E uses histograms for data representation. The explanation of the statistical asterisks was added to the legend l521 (P-value : * p<0.05, **: p<0.01).
All figure legends: Please list all abbreviations used in each figure in the respective legend.	We extensively reviewed the abbreviations and listed them at the end of the legends.

Reviewer #3 (Remarks to the Author)

Summary: The authors demonstrate, as a proof of principle, the application of tissue clearing and light sheet fluorescence microscopy (LSFM) towards the imaging and analysis of retinal and hyaloid vessel development. The authors provide beautiful and high resolution light sheet acquired images of developing eyes and demonstrate first proof of principle utility for using this approach to study hyaloid vessel formation. The authors also demonstrate a proof of principle application for studying branch retinal vein occlusion (BRVO) models.

There are several limitations to this manuscript, including, the already demonstrated the utility of tissue clearing and light sheet microscopy towards studying ocular vasculature from several groups (Henning et al., 2019, Vigouroux et al, 2020; Ye et al., 2020, Prahst et al 2020, Gurdita et al., 2020), so it is unclear what major advances this study provides to the field.	We agree that we should have provided more elements of comparison with the achievements of other teams, which would have better highlighted the specific interest of our research. As pointed by Reviewer 2, Henning et al. published the first images of LFSM of the adult choroidal and retinal mouse vessels after clearing, using a slightly different protocol called EyeCi, which stems from the iDISCO+ procedure that we reported (Renier et al, 2016). Yet, Henning e al. did not provide any quantitative data and no developmental or pathological data. In the following years, several papers added their contribution to the understanding of the
--	--

	contribution of LSM to ocular histology. -Vigouroux et al (2020) described developmental abnormalities related to the DCC gene defect, using a similar approach to ours (yet not observing nor quantifying blood vessels and microvasculature).-Yang et al. (2020) provided essentially data on the external vascularisation of the anterior segment of the eye.-Ye et al (2020). provided essentially a proof of concept of the feasibility of LFSM in pigmented eyes, but did not provide in-depth description of their observations, hence bringing little contribution to the knowledge of ocular physiopathology.-Gurdita et al. (2021) provided morphometric data of the retinal and choroidal volume, and spatially characterized grafted photoreceptors. Again, no details of the organization of the microvasculature were provided.-Prahst et al (2020). provided a detailed comparison of confocal microscopy and LFSM. However, they provide no data of the eye in toto; indeed, all retinal samples were dissected from the choroid and sclera. By comparison with the abovementioned papers, we provide in our paper the following:1-a more detailed, quantitative description of the different vascular layers at the microscopic level over a large field (i.e. the eye in toto) together with fine, multilayer numerical dissection, showing in particular the connections between anterior and posterior segments circulations2- a comprehensive, quantitative, longitudinal overview of the hyaloid vasculature, showing that the interface of anterior and posterior segments maintained a high capillary density late in development.3-the identification of novel features such as contacts between hyaloid vessels and the peripheral retina4-The study of vascular environnement in toto, with perivascular cells such as pericytes, astrocytes, and hyalocytes in their 3D organization through development. Their crucial role for a functional, physiological vascular network is often explored through classical histological means, but has not been described in toto in correlation to vascular development.
--	---

	We added this complementary information in the discussion.
Similarly, although a correlation between OCT and LFSM images in the BRVO model are demonstrated, it is unclear what new insights this provides.	In order to comply with this remarks, we modified the paragraph in the discussion section the following way: BRVO animal model had proven complex to document in the past using classical histology (Paques, Roubeyx). To our knowledge, we provide here the first spatially-preserved reconstruction of the pattern of collateral vein development following BRVO. LFSM prevents the inadvertent sectioning of these collateral vessels that may occur when flat-mounting the retina, and also is much faster. Figure 5 shows that establishing a correspondence between the in vivo imaging by SLO and histology by LFSM was possible and ensured proper identification of a collateral vessel that may otherwise be difficult to detect within the dense mesh of retinal vessels post-mortem. The dilated collateral channels could be followed in 3D through the entire retina and all vascular layers and studied from all angles, while also imaging the occlusion site and the entire vascular environment around the post-occlusive remodeling. This enabled it to detect affected veins in distally located areas, in particular in the extreme periphery of the retina. Indeed, a single branch retinal vein occlusion affects roughly half of the vasculature because of the deviation of venous flow, which finds its way out through adjacent channels (which also runs along the extreme periphery of the retina). Even more, it shows that LFSM enables better documentation of the extent of venous dilation than in vivo imaging. It also demonstrates that the vascular dilation and tortuosity is preserved to some extent during tissue processing. This approach will offer a powerful tool for anatomical studies, pharmaceutical screening, or mutant mouse lines characterization.

Major Concerns:

It is unclear how the findings in this study differ, compare or add to the existing understanding of retinal vascularization and hyaloid development using traditional approaches ie. serial reconstruction or wholemount.

Serial reconstruction from parallel sections of the spherical eye cannot by definition provide an appropriate *in toto* view; additionally, missing sections cause gaps in capillary continuity. The technique is also highly time consuming, making any comparative approach complicated and cumbersome. Wholemount approach for hyaloid vessels does not preserve at all their spatial organization, their connections and the network real complexity. Flatmounted retinas also present their own histological bias, the main one been the obligatory sections through the tissue to obtain the “star-shaped” flatmounted retina. While studying structures such as the vascular network, this level of tissue-destruction biases the interpretation of the organization of the network. Comparison to *in vivo* imaging is also challenging when using flatmounted tissue, since the spatial organization is lost.

We believe that the approach we used enable us to document several new aspects relative to ocular vasculature because of the preserved spatial disposition of the delicate, gel-like vitreous structures: Documenting morphometric characteristics of the hyaloid development with minimal spatial disturbance enabled fine quantitative analysis, which showed for instance that there was no evidence of postnatal increase of vascular density. *In toto* viewing also allowed detailing the connections between the anterior and posterior circulations, which is seldom explored and by definition inaccessible by flatmounts. For the first time we could characterize the density of vessels throughout the hyaloid system, showing that the peak capillary density remains around the lens.

The study lacks a comparison to existing tissue clearing and LSM approaches which are now well documented. Thus, the paper would be improved by using this approach to general novel insights in hyaloid development/defects and BRVO models.	We agree that our paper should set its focus more on the novel insights provided relative to other clearing approaches. To our knowledge it is the first documentation in such fine details of the hyaloid system. We could indeed document the capillary density and also identify for the first time drainage routes of the hyaloid system. The high precision images provided here indeed open the path toward identification and mapping of focal or patchy damage to ocular vessels, such as shown in the BRVO case; it is indeed to the best of our knowledge the first documentation of the 3D disposition of an entire collateral vessel. This has been specified in various places in the introduction and conclusion.
--	---

Minor problems and concerns:

Line 82: Duplicated paragraph	This has been corrected
Line 119: If albino eyes were used then why was bleaching necessary?	In-toto eyes imaging requires a complete depigmentation even of albino eyes, as residual blood cells can significantly impair imaging; also, paraformaldehyde-fixed albino tissues are translucent but not transparent enough for LFSM.
Line 130: Can direction of blood flow be inferred from static images? Is this based on anatomy alone? If this is common knowledge in the field can the authors clarify this to make the manuscript more accessible to a non-expert.	The arrow in figure 1E shows vessels that have been shown to anatomically correspond to choroidal drainage identified in vivo. A reference (Paques et al. 2007) has been added.
Line 153: Confusing to mention NG2 staining here because it is not discussed until a later section.	This was corrected.
Line 168: What does 'numerically' dissected mean?	We modified the term to "virtually". We used the 'volume' tool in Imaris to select the structures of interest on the images and isolate them from the rest to be able to see them separately in 3D. Thus the use of 'numerically' or 'virtually' dissected.

Line 179: Supplemental Figure 4, why is the CD31 stain so saturated?	While observing samples, it was incidentally discovered that the very slight background noise of samples allowed for the structural observation of the vitreous. Since there isn't a lot of background noise, to have a clear image the intensity of the entire image had to be increased, explaining the saturated CD31.
Line 182: Supplemental Figure 5, it would be useful to have a low magnification image here to help spatially orient the reader.	We added a panel with an orienting image.
Line 183: Can one draw any conclusions about function with a static IHC image in this context?	We agree that the sentence line 183 is unclear. By function, we meant that these capillaries connecting the hyaloid system to the retina were not perfusing retinal vessels. We corrected the sentence the following way: "We observed that these vessels connecting the hyaloid system to the retina formed a loop within the hyaloid system, i.e. were not perfusing the retina. Therefore they were not contributing to the perfusion of retinal vessels."
Line 192: Indicate figure.	This has been done
Line 213: Is this the correct figure reference? If it's supposed to be 3B then it is also not cited in the correct order in the paragraph.	The correct reference is 'Figure 4B, insert'. We added 'Figure 4B, insert shows an example of a remnant capillary with a pearl string-like appearance and no CD31 labeling' for clarification purposes, l 184.
Line 226: P0 instead of P6.	This has been done
Line 228: Point the discontinuous NG2 labelling more clearly in the figure. Are you referring to the green vessel only in the inset?	Yes. We changed the sentence to make that fact clearer.
Line 240: The figure 4F is cited out of order	This has been corrected.
Line 254: It's not appropriate to use 'back and forth' since this analysis has not been demonstrated to be bidirectional.	This has been corrected

Line 257: Duplicated section.	This has been corrected
Line 273: Should acknowledge that this is not the first attempt at imaging vessels at the capillary level and should compare to existing cleared eye literature and state the advance.	We agree that we should have provided more elements of comparison with the achievements of other teams, which would have better highlighted the specific interest of our research. As pointed by Reviewer 2, Henning et al. published the first images of LFSM of the adult choroidal and retinal mouse vessels after clearing, using a slightly different protocol called EyeCi, which stems from the iDISCO+ procedure that we reported (Renier et al, 2016 and Darche et al, ARVO poster 2016). Yet, Henning e al. did not provide any quantitative data and no developmental or pathological data. In the following years, several papers added their contribution to the understanding of the contribution of LFSM to ocular histology. -Vigouroux et al (2020) described developmental abnormalities related to the DCC gene defect, using a similar approach to ours (yet not observing nor quantifying the microvasculature). -Yang et al. (2020) provided essentially data on the external vascularisation of the anterior segment of the eye. -Ye et al (2020). provided essentially a proof of concept of the feasibility of LFSM in pigmented eyes, but did not provide in-depth description of their observations, hence bringing little contribution to the knowledge of ocular physiopathology. -Gurdita et al. (2021) provided morphometric data of the retinal and choroidal volume, and spatially characterized grafted photoreceptors. Again, no details of the organization of the microvasculature were provided. -Prahst et al (2020). provided a detailed comparison of confocal microscopy and LFSM. However, they provide no data of the eye in toto; indeed, all retinal samples were dissected from the choroid and sclera. By comparison with the abovementioned papers, we provide in our paper the following: 1-a more detailed, quantitative description of the different vascular layers at the microscopic level over a large field (i.e. the eye in toto) together with fine, multilayer numerical dissection, showing in particular

	the connections between anterior and posterior segments circulations 2- a comprehensive, quantitative, longitudinal overview of the hyaloid vasculature, showing that the interface of anterior and posterior segments maintained a high capillary density late in development. 3-the identification of novel features such as contacts between hyaloid vessels and the peripheral retina 4-The study of vascular environment in toto, with perivascular cells such as pericytes, astrocytes, and hyalocytes in their 3D organization through development. Their crucial role for a functional, physiological vascular network is often explored through classical histological means, but has not been described in toto in correlation to vascular development.
Line 315: Confocal microscopy images were not provided.	We deleted the sentence.
Line 369: Statements such as ‘up to 10 days for primary antibody incubation’ should be clarified so that readers may be able to easily reproduce this approach. At the very least, a minimum antibody incubation time should be provided. Alternatively, if incubation times differed depending on the primary antibody, then each primary antibody incubation could be specified. These primary and secondary antibody incubation times are also starkly different from existing eye clearing protocols. Can the authors comment on how they settled on these incubation times?	We use a protocol based on the protocol described by Belle et al, 2017, and as indicated in the paper, incubation with primary antibody can go from “7 to 14 days depending on tissue size and density”. Through trial and error, it was determined that optimal labeling could be reached in this time period on the entire eye. A minimum of 7 days has to be respected, and has been used on the smaller tissues of developing eyes. Adult eyes, bigger in size, are incubated for 10 days. This has been better described in the paper’s methods for better reproducibility. I332
Line 376: Why wasn’t EyeDISCO used for the clearing procedure? It may be helpful to acknowledge differences between this papers use of the standard iDISCO and the optimized EyeDISCO protocols.	EyeDISCO and our protocol were developed by different teams in our institution, and most of our study results were already obtained when EyeDISCO was published. The EyeDISCO protocol is a technical progress dedicated to the study of pigmented eyes, whereas this project centered around a holistic, anatomical description of vascular networks in the eye through the entire development of the mouse. To this end, albino mice were used, thus necessitating few modifications to the trusted and peer approved protocol of iDISCO + described in Belle et al, 2017,

	since no heavy depigmentation was needed. H2O2 depigmentation presenting risks for tissue damage, development of a stronger depigmentation protocol was not needed for our objective.
Line 418: Why were non-parametric statistical tests used? Were parametric test assumptions violated?	Non-parametric tests were used as we could not test normality given the small sample size (between 3 to 6 values, plotted individually on the graph).
Line 551: Panel D, label Y axis in figure	This was done.
Line 555: Panel E, label on Y axis instead of X axis. Combine graphs on one page to use legend once.	For clarity of reading, we switch the label in a title position.

REVIEWERS' COMMENTS:

Reviewer #2 (Remarks to the Author):

The authors have addressed all my comments and questions, I am happy to recommend this manuscript for publication.

There are two things I would like the authors to address still:

1. There seem to be parts on the filters missing in the methods section. (§ filters)
2. There is no histogram shown in Fig3. These are bar plots which show mean values and deviations whereas histograms show frequency distributions.

Reviewer #3 (Remarks to the Author):

The authors have done a great job in addressing our comments.

We thank the reviewers for their kind comments.

The two issues concerning remaining precision on the imaging set up and the semantical correction have been addressed in the updated manuscript.

Best regards,

Michel Paques